# Scar Management in Pediatric Patients

**DOI:** 10.3390/medicina61040553

**Published:** 2025-03-21

**Authors:** Sydney Barone, Eric Bao, Stephanie Rothberg, Jose F. Palacios, Isabelle T. Smith, Neil Tanna, Nicholas Bastidas

**Affiliations:** 1Division of Plastic and Reconstructive Surgery, Lewis Katz School of Medicine, Temple University, Philadelphia, PA 19140, USA; 2Division of Plastic and Reconstructive Surgery, Northwell Health, New York, NY 10022, USA; eric.bao120@gmail.com (E.B.); isabelletsmith@gmail.com (I.T.S.);; 3Division of Plastic and Reconstructive Surgery, Donald & Barbara Zucker School of Medicine, Hofstra University/Northwell, Hempstead, NY 11549, USA

**Keywords:** pediatrics, pediatric scars, scars, scar management

## Abstract

*Background and Objectives*: Pediatric patients can acquire scars from both accidental injury and surgical procedures. While scars cannot be avoided if a full-thickness injury occurs, scar visibility may be minimized through a variety of approaches. In this narrative review, we evaluate the current evidence and propose an algorithm for scar management in pediatric patients. *Materials and Methods*: A review of the literature was performed for scar management techniques for pediatric patients. Management modalities based on the type of scar and dosing, treatment regimen, and safety profiles are described in this article and used to create a scar management algorithm. *Results*: The initial step to scar management in the pediatric population involves ensuring minimal wound tension, which can be achieved through making the incision along relaxed skin tension lines, and early, minimal tension wound closure. Subsequent treatments to optimize scar care should begin 2–3 weeks following wound closure and involve the application of silicone gel or sheets and scar massaging. When topical products are insufficient, laser therapy can be utilized for the management of immature erythematous or thick scars. When mature, pathological scars form such as atrophic scars, hyperpigmentation, hypertrophic scars, or keloids, a combination of modalities is recommended. These modalities vary by scar type and include retinoids and dermabrasion for atrophic scars; retinoids, hydroquinone, and laser therapy for hyperpigmentation; and pressure therapy, corticosteroids, and laser therapy for hypertrophic scars and keloids. When mature, pathological scars persist following 12 months of non-invasive therapies, surgical excision should be considered. *Conclusions*: Several treatment options are available to manage scars in the pediatric population depending on scar type.

## 1. Introduction

It is universally accepted that all full-thickness surgical incisions result in the formation of a scar [1]. However, all scars are not the same. Immature scars, typically less than one year old, can become erythematous or thickened. These scars occur soon after initial injury and are predominantly composed of inflammatory cells. Mature scars are typically older than one year old, and in ideal cases, heal as thin pencil lines with minimal color difference between the scar and areas of surrounding skin (Figure 1) [2]. In less ideal cases, atrophic scars, hyperpigmentation, hypertrophic scars, or keloids may result. While no outcome can be guaranteed and multiple factors influence scar cosmesis, strict adherence to evidence-based principles may help surgeons minimize suboptimal scar appearance in their patients [3]. Scar cosmesis must be especially maximized in children as these scars will be present as they form their self-perceptions and begin socializing with others. Scars in children also grow proportionally to the child, becoming more prominent and unsightly as they age. In this narrative review, we propose an evidence-based algorithm for the management of various scar types from incision planning to last-resort scar revision surgery in the pediatric population.

## 2. Methods

An electronic-based search was conducted using the following databases: PubMed, Google Scholar, and UpToDate. The following terms and their combinations were used: “scar management”, “pediatric scars”, “scar cosmesis”, “immature scars”, “atrophic scar”, “hyperpigmentation”, “hypertrophic scar”, “keloid”, “incision planning”, “wound closure”, “topical scar treatment”, “silicone gel”, “scar massage”, “laser therapy”, “intralesional corticosteroids”, and “scar revision”. The included articles evaluated acne, post-operative, and post-burn scars with less emphasis on post-traumatic scars. Articles published from 2006 to 2024 were included. Randomized controlled trials, case series, systematic reviews, meta-analyses, and electronic textbook chapters were included. The included articles were analyzed to generate an evidence-based scar management algorithm.

## 3. Managing the Scar Before It Forms

### 3.1. Incision Planning

The cornerstone of scar management is incision planning. While the surgeon will not always have complete control over where an incision is placed, as is the case in an unplanned laceration, pre-established principles should be honored as much as possible to optimize scar cosmesis. Tension on wound closures is known to result in stretched, hypertrophic, and poor scar formation [4]. Therefore, when possible, incision placement should ensure minimal wound tension [3,5,6]. This can be accomplished by following relaxed skin tension lines (RSTLs) which are theoretical lines where skin tends to furrow with local muscle contraction [5]. Incisions should run parallel to RSTLs when feasible to minimize tension on the wound [5,7]. Incisions in the scalp should be beveled so that hair follicles can later grow through the incision to disguise the scar [7].

### 3.2. Wound Closure

Similarly to incision planning, wound closure should also minimize wound tension [5]. Early primary wound closure is critical to induce wound epithelialization as soon as possible because wound epithelialization delayed beyond 10–14 days increases the risk for the formation of hypertrophic scars [5]. Full-thickness incisions should be closed in layers starting with the deep dermal layer and then the epidermal layer. Deep dermal sutures decrease tension in the superficial layers and align the dermis for better cosmetic results [8,9]. Wounds closed in a single deep dermal layer had more prominent appearances at 3 months and more noticeable scar color at 12 months compared to wounds closed in two layers [10]. Full-thickness wounds closed in a single epidermal layer may result in “parallel rows” as seen on a railroad track due to pressure necrosis of the skin underneath the suture [11]. However, if the suture is tied too loosely, wound dehiscence may occur.

No statistically significant differences in scarring were reported between the use of resorbable or non-resorbable sutures [12,13]. However, patients reported less discomfort with resorbable sutures, which is beneficial when treating pediatric patients [12]. Resorbable sutures should be used for deep dermal suture layers. Non-resorbable sutures require strict patient follow-up for suture removal to prevent punctate scarring formation. As a result, resorbable sutures have the added benefit of decreasing the need for patient follow-up and punctate scar formation. The smallest sutures that can adequately overcome the wound tension should be utilized and if non-resorbable sutures are used, they should be removed as soon as the wound can hold itself together [5]. For the cutaneous closure of facial scars, running subcuticular, fine running, or interrupted sutures have been recommended, though superior cosmetic results with continuous subcutaneous sutures compared to interrupted sutures, have been reported [7,14].

## 4. Immature Scars

Immature scars appear following suture removal. Therapies such as silicone gel or sheeting and scar massaging should begin 2–3 weeks following the formation of the scar to improve cosmetic outcomes and prevent the formation of mature, pathologic scars. Silicone gel and sheeting have the most evidence-based support as effective topical modalities for the improvement of scar cosmesis, and the prevention and treatment of pathologic scars (Figure 2) [15,16,17,18,19,20,21,22]. Silicone should ideally be applied at least 12 h per day beginning 2 weeks post-operatively [23,24,25]. Data vary regarding the length of time silicone should be utilized, ranging from 2 to 6 months [25,26]. Silicone gel can decrease scar volume and increase wound elasticity in 60–100% of the cases [27].

Scar massaging is an additional modality that should be utilized with immature scars as it improves post-surgical scar appearance, yielding a 90% improvement in scar appearance or Patient Observer Scar Assessment Scale [28]. Scar massaging should begin 2–3 weeks post-operatively, but should be delayed further if the wound has not completely closed [25]. Data on the suggested massage times are weak and range from 10 min two times per day to 30 min two times per week [28,29,30]. This should be performed for at least 6 weeks and studies report its use upwards to 6 months in combination with standard therapies such as silicone gel [25,28]. However, massage therapy remains a cost-effective and harmless method of scar management with added psychological and anxiolytic benefits for the patients and family.

Despite the use of silicone and massage therapy, immature scars sometimes remain erythematous and/or thickened. In this case, laser therapy should be added to the treatment regimen. Non-ablative lasers include pulsed dye laser (PDL), neodymium-doped yttrium aluminum garnet (Nd:YAG), and 1450 nm diode laser, whereas ablative lasers include CO_2_, argon, and Erbium:YAG (Er:YAG) lasers [31,32]. Non-ablative lasers, especially non-ablative fractional lasers, have fewer adverse effects, but require more treatment sessions than ablative lasers [29,33]. Also, scar pigmentation has been shown to be significantly more improved with the use of non-ablative PDL compared to CO_2_ ablative fractional laser but scar height is significantly improved with the CO_2_ laser [34]. Other studies indicate that fractional ablative lasers CO_2_ 10,600 nm and Er:YAG 2940 nm demonstrate the best results regarding both scar erythema and height [35]. PDL can be started as soon as suture removal or within one month of surgery to optimize scar appearance. It is often performed every 3–4 weeks for 4–6 sessions [36,37]. CO_2_ laser, however, should be initiated within 6–10 weeks following scar formation though the number and frequency of treatments is case-dependent [31]. Importantly, fractional ablative CO_2_ laser resurfacing of traumatic and surgical scars is safe, well tolerated, and efficacious in pediatric patients [38,39,40].

## 5. Mature Scars

During the ideal transition from immature to mature scar, inflammatory cells, endothelial cells, and a majority of the fibroblasts undergo apoptosis and a band of collagen fibers remains [41]. Minimizing wound tension and utilizing silicone gel or sheets, massage therapy, and sometimes laser therapy help ensure a positive cosmetic outcome as the mature scar forms. However, despite these therapies, the transition from immature to mature scar can be aberrant, resulting in atrophic scars, hyperpigmentation, hypertrophic scars, or keloids. These pathological scars can significantly impact the functional and psychosocial quality of life of pediatric patients. As such, it is important to address and manage these pathological scars.

### 5.1. Atrophic

Atrophic scars occur from a loss of collagen, elastin, and connective tissue in the dermis, resulting in a depression of the skin [42]. Dermatologic conditions, such as acne, are the most common causes, rather than intentional incisions such as from surgery [43]. Pediatric patients should be of particular focus for atrophic scar management as they are significantly more likely to develop atrophic scars [43].

Atrophic scars are classified into three types: ice pick, boxcar, and rolling, which inform treatment decisions (Figure 3) [44]. Depending on the scar severity and patient preference, topical retinoids may be used as a non-invasive, cost-effective approach. Topical retinoids have demonstrated improvement both as a monotherapy and in combination with other treatment modalities. The daily use of adapalene 0.3% gel improved skin texture and atrophic scars within 24 weeks by 50% and over 80% per investigators and subjects, respectively [45]. A more recent study showed daily Trifarotene, one of the newest FDA-approved topical retinoids, demonstrated significant improvement compared to vehicle-treated scars [46]. For combination treatment, daily adapalene 0.3% and benzoyl peroxide 2.5% significantly decreased the total scar count [47]. The youngest patient included in these studies was 16, so additional considerations may be necessary for younger pediatric patients, but studies have demonstrated that it is safe to use topical retinoids and benzoyl peroxide for acne management of pediatric patients [48].

Chemical peels are another treatment most effective for boxcar atrophic scars, and they include glycolic acid, Jessner Solution, pyruvic acid, salicylic acid, and trichloroacetic acid. The strength of the acid may be tailored to the depth of skin penetration desired. For glycolic acid, concentration can vary from 30 to 70% for chemical peels, with more tissue damage occurring at higher concentrations [44,49]. For best results with atrophic scars, treatment with 70% glycolic acid biweekly for five sessions is recommended [44]. The face should be prepped with alcohol, and then the solution applied and allowed to sit for 3 to 5 min before rinsing off with water and applying a moisturizer [50,51]. Salicylic acid is most effective at 30% concentration with treatment every 3 to 4 weeks, totaling 3 to 5 sessions [44]. The treatment should be conducted similarly to glycolic acid peels [52]. Jessner Solution may be used as a light peel or to prepare the skin for a subsequent TCA peel [53]. In total, 1 to 3 coats of the solution yield a very superficial peel, 4 to 10 coats are used for a superficial peel, and greater than 10 coats are needed for moderate penetration. No studies have assessed the safety of Jessner Solution in pediatric patients; however, use is contraindicated in patients that had abnormal wound healing, active inflammation, dermatitis, infection, or used isotretinoin within the last 6 months [44]. Pyruvic acid is used as a peel at 40 to 70% concentration [54]. Lastly, trichloroacetic acid (TCA) may be used for a chemical peel at 10 to 35% concentration. However, 50 to 70% TCA may be applied via the CROSS method to deliver highly concentrated TCA directly to the scar (Figure 4) [44,55]. For the CROSS method, the face should be cleaned with soap and water, and then alcohol. The solution may be applied to the scar with a fine pointed applicator, such as a toothpick, until frost appears. Frequency may vary, with sessions occurring every 2 to 4 weeks for a total of 3 to 4 treatments [44,55]. The CROSS method has been utilized on adolescents as young as 15 years, demonstrating safety in older pediatric patients [55]. Following any chemical peel, patients should use broad-spectrum sunscreen daily for a minimum of 2 to 3 months [56].

Laser therapy offers another non-invasive treatment modality for boxcar and rolling atrophic scars (Figure 5) [57]. Importantly, both non-ablative and ablative laser therapy are safe for the management of atrophic scars in the pediatric population [38,39]. For non-ablative laser, 3 to 5 treatments occurring every 3 to 4 weeks are standard, with 51% to 75% improvement in 87% of patients after 3 treatments [58]. Scar texture can also significantly improve after three treatments [59]. However, additional sessions may be required for scars resulting from more traumatic modalities, such as burns [60]. Ablative laser treatments occur every 3 to 6 weeks, with around 3 treatments needed. This modality may be better suited for post-burn and post-traumatic atrophic scars as up to 70% of the patients report good or excellent responses, with 68% of the post-acne scar patients also reporting excellent responses [61]. Combination treatment with ablative and non-ablative lasers demonstrated greater improvement with fewer complications than ablative treatment alone [62]. Such treatments include topical poly-L-lactic acid, PRP, and punch elevation or subcision, with treatment regimen determined on a case-by-case basis [63,64,65,66].

Dermabrasion and microdermabrasion are other minimally invasive treatments for atrophic scars, particularly rolling scars [67]. Both treatments make numerous, controlled small punctures to the skin to disturb the dermal collagen connected to the scar tissue [68]. This triggers wound healing, including collagen production and remodeling, thickens the epidermis and dermis, and increases elasticity [68]. Microdermabrasion may be best suited for pediatric patients as the procedure is painless, does not require general anesthesia, and has less severe complications [44]. However, dermabrasion is more effective, particularly for deeper scars [68]. Sessions may be performed every 2 to 8 weeks, with at least 4 to 6 treatments [68,69,70]. Human atrapid insulin at a concentration of 40 IU/L or PRP may be applied immediately following the microdermabrasion and must sit for 30 min to 1 h [67,71]. Chemical peels, such as Jessner Solution and 70% glycolic acid peel, significantly improved results compared to microdermabrasion alone [72,73].

Autologous fat grafting is a more invasive, but a routine and safe procedure that uses harvested fat to fill in the depression of the atrophic scar. Adipose-derived stem cells can regenerate the dermis and subcutaneous tissue as well as stimulate new collagen formation to improve the appearance of scars [74]. Autologous fat grafts can significantly improve the appearance of atrophic scars as soon as 3 months post-operation, particularly for the forehead and less for the nose, infraorbital region, and chin [75,76]. Importantly, autologous fat grafting has been safely utilized in the pediatric population [74,76]. Fat grafts may also be combined with condensed nanofat to elevate the surface and base of the scar, which can significantly improve the color, stiffness, thickness, irregularity, and pliability of the scar [77].

### 5.2. Hyperpigmented

Hyperpigmented scars, also known as post-inflammatory hyperpigmentation (PIH), occur when a skin defect becomes inflamed, causing an overproduction of melanin. Patients with darker skin naturally produce more melanin, so they are more affected by PIH. PIH can last several months to years, which can damage the pediatric patient’s self-image and confidence.

Treating PIH effectively oftentimes involves treating the underlying cause. Most instances of PIH in the pediatric population are caused by acne vulgaris, though other common causes include atopic dermatitis and impetigo [78,79]. Therefore, utilizing a daily gentle skin care routine that includes cleansers and moisturizers goes a long way in preventing PIH. Additionally, photoprotection using clothing or sunscreen with a sun protection factor (SPF) greater than or equal to 30 is recommended to reduce inflammation of the skin and reduce PIH [80]. Sunscreen should be applied before any contact with sunlight and re-applied after 2 h or after water-based activities [78].

Topical retinoids applied once daily are the first-line therapy for PIH caused by acne (Figure 6) [81,82,83]. Benzoyl peroxide, azelaic acid, or superficial chemical peels such as glycolic acid or salicylic acid, all of which are safe in pediatric patients, may be used alone or in combination with topical retinoids to further improve acne-induced PIH [48,81]. Of note, many studies on the treatment of acne-induced PIH have been conducted on individuals with darker skin, so effectiveness may vary for those with lighter skin tones.

Epidermal PIH presents as light to dark brown spots. The first line treatment agent is hydroquinone, though a triple combination therapy of 5% hydroquinone, 0.1% tretinoin, and 0.1% dexamethasone is used to improve efficacy and reduce side effects of hydroquinone [84,85]. This triple therapy, known as the Kligman formula, is to be used once daily for 8 weeks once PIH has developed. There is scarce literature detailing the efficacy and adverse effects of hydroquinone in children under 13 years old, but it is safe for those older than 13. Similarly, tretinoin and topical corticosteroids in low doses and short intervals are well tolerated in pediatric populations [86,87]. In cases when hydroquinone is not accessible or the side effects outweigh the benefits, other topical agents can be used. Retinoids, specifically tazarotene, can be used once daily for 12–18 weeks, soy moisturizer twice daily for 12 weeks, 4% niacinamide once daily for 9 weeks, 2% N-acetyl glucosamine twice daily for 8 weeks, kojic acid plus emblica and glycolic acid for 12 weeks, or 15% azelaic acid twice daily for 16 weeks [88,89,90,91,92,93,94]. Although there is a paucity of literature supporting the use of N-acetyl glucosamine or kojic acid in the pediatric population, glycolic acid and azelaic acid have previously been studied and used safely for children [95,96].

Dermal PIH appears as blue-gray or black areas of discoloration and can be treated with 1064 nm QS Nd:YAG laser with low fluence or 1550 nm erbium-doped laser. The treatment frequency for the Nd:YAG laser is not clarified, but shows promising results compared to other lasers [97]. The Nd:YAG non-ablative laser is safe for use in pediatric patients. The erbium-doped laser is generally reserved for cases of PIH refractory to first-line agents and has yielded excellent results [98,99,100].

### 5.3. Hypertrophic

Hypertrophic scars are pathological scars that result from excessive inflammatory mediators and fibroblast proliferation during scar formation, but remain within the margins of the initial wound (Figure 7) [2,41,101,102]. Hypertrophic scars sometimes regress within the first year following formation, but nonetheless, they remain a challenge to treat.

In patients with a history of pathological scar formation, a wound in a high skin tension area, or if a scar begins to appear hypertrophic, pressure therapy is recommended. Pressure therapy utilizes the help of devices or garments, such as compression dressings, pressure buttons, or ear clips, to induce local ischemia, increasing collagenase activity and accelerating wound healing [27,103]. Pressure should be maintained between 20 and 30 mmHg for 18–24 h per day for at least 4–6 months, but ideally upward to 2 years [27]. A meta-analysis of 12 randomized controlled trials reported an improvement in thickness, color, and hardness of hypertrophic scars secondary to burn injury with pressure maintained at 15–25 mmHg for 2–12 months [104]. Patient compliance, especially in the pediatric population, may be hindered by the prolonged time requirement of pressure therapy or by the uncomfortable nature and unsightly appearance of pressure garments.

Intralesional corticosteroid injections should be added to the treatment regimen if less invasive modalities including silicone gel or sheets and pressure therapy are not effective alone after 2 months [105]. Corticosteroids decrease inflammatory mediators, inhibit keratinocyte and fibroblast proliferation, and induce vasoconstriction which reduces the nourishment of scar tissue [29,106]. Intralesional corticosteroids can induce 50–100% regression of hypertrophic scars [3,22]. The recommended dose ranges from 2.5 to 40 mg/site of Triamcinolone Acetonide (TAC) mixed with local anesthetic [3,24]. There is weak evidence dictating the length of follow-up or number of treatments for steroid injections in the pediatric population, with sources suggesting a case-by-case basis [39,107]. TAC injections can be painful, leading to the loss of follow-up in pediatric populations, though this can be partially remedied by topical anesthetics.

Other intralesional therapies that may be utilized for the treatment of hypertrophic scars include 5-fluoruracil (5-FU) or bleomycin. 5-FU is a pyrimidine analog that inhibits fibroblast proliferation [27]. 5-FU can be used as monotherapy but is more efficacious when used in combination with TAC [108]. 5-FU in combination with TAC improves hypertrophic scar height, erythema, and patient satisfaction significantly more than TAC alone [109,110]. Monthly administration of 0.9 mL of a 50 mg/mL solution of 5-FU combined with corticosteroids has been recommended for severe hypertrophic scars [105]. Bleomycin is a cytotoxic antibiotic that inhibits collagen synthesis and induces apoptosis [27]. Intralesional bleomycin monotherapy has been reported as more effective than intralesional TAC alone, intralesional 5-FU alone, and combined intralesional TAC and 5-FU in the treatment of hypertrophic scars [111,112]. A total of 1.5 IU/mL of bleomycin should be administered through multiple intralesional injections [105]. However, it is questionable if pediatric patients are good candidates for intralesional 5-FU and bleomycin therapy as there is limited literature regarding the safety of these chemotherapeutic agents in the pediatric population outside of cancer treatment. On the other hand, topical TAC and 5-FU are safe in the pediatric population with topical thermal ablation immediately prior to the topical application of TAC and 5-FU enhancing the delivery of these therapies [113].

Laser therapy is efficacious for treating hypertrophic scars and should be utilized in combination with previously mentioned therapies as it may enhance the delivery of adjuvant scar treatments (Figure 8) [114]. PDL and CO_2_ ablative laser have been used alone or in combination for the treatment of hypertrophic scars [115]. The clinical efficacy of PDL in the treatment of hypertrophic scars is widely accepted, with reports of reduction in scar pruritus, erythema, scar height, and skin surface texture with its use [27,32,116]. The optimal reported time interval between laser treatments varies from 4 to 6 weeks to 2–3 months, with most studies suggesting six weeks as the optimal interval [117]. Some studies suggest that once the erythema, hyperemia, and pruritus responses plateau with PDL, a fractional CO_2_ laser can then be initiated to improve hypertrophic scar texture and pliability [117,118]. One study in children noted significant improvement in quality, pain, and pruritus of hypertrophic scars following treatment with a combination of CO_2_ laser and PDL [119].

When the above therapies have been utilized for at least 12 months for primary wound treatment and the hypertrophic scar has failed to regress, surgical re-excision is recommended [4]. The best way to prevent the recurrence of hypertrophic scar is to minimize wound tension with wound planning and closure.

### 5.4. Keloids

Keloids are another form of pathological scarring that results from the dermal proliferation of fibrous tissue. Unlike hypertrophic scars, keloids grow beyond the margins of the initial wound (Figure 9) [41]. The management of keloids is similar to the management of hypertrophic scars, but should be more aggressive as keloids are less likely to regress and can continue to grow for a longer period of time than hypertrophic scars.

Similarly to hypertrophic scars, pressure therapy should be utilized for keloids. If a patient has a history of keloids, pressure therapy should be utilized prophylactically. Pressure therapy is unlikely to resolve keloids alone but should be used in combination with other modalities including silicone gel or sheets, intralesional corticosteroids, and/or surgical excision [120].

Intralesional corticosteroids are a first-line therapy for the treatment of keloids and the best results are reported when intralesional corticosteroids are used in combination with other treatment modalities such as compression, radiation, laser, or surgery [24,106,121]. Radiotherapy is not widely accepted in the treatment of keloids in pediatric patients due to its negative effects on growing tissues and the risk of secondary malignancy [2,122,123]. In the pediatric population, intralesional TAC injections alone resulted in an 82.7% reduction in keloid size, and intralesional TAC utilized in combination with surgical excision and CO_2_ laser therapy resulted in complete clearance of keloid in 82.4% of the patients [40,122,124].

If the keloid has not improved after 8–12 weeks of conservative therapies and intralesional TAC alone, the addition of 5-FU should be considered [105]. 5-FU is considered an “off-label” use for scar management [39]. Low dose 5-FU at a concentration of 1.5–5 mg/mL has been proposed for the treatment of keloids and prevention of keloid recurrence as higher concentrations are associated with more side effects including keloid necrosis and ulceration [125]. 5-FU is more effective for treating keloids when used in combination with intralesional TAC compared to when used as monotherapy and it is recommended to combine 1.5–5 mg/mL of 5-FU with 3–9 mg/mL of steroid [109,125]. Injections should begin at 4-week intervals for a few months, followed by injections every 6–10 weeks for several months, and lastly adjusted to every 12 weeks [125]. Regarding intralesional bleomycin, intralesional doses of 1.5 IU/mL are recommended [105]. In a retrospective study of 314 patients, surgical shave excision of keloids was performed and after re-epithelialization, bleomycin injections were given monthly until pain and/or pruritis associated with the keloid resolved and the keloid disappeared [126]. In this study, 87% of the patients were very satisfied with the complete flattening of the keloid, 11% of the patients were moderately satisfied with the significant flattening of the keloid, pain and pruritis were resolved in 100% of the patients, and 2% of the patients had keloid recurrence [126]. The use of both 5-FU and bleomycin for scar management has not been specifically studied in the pediatric population.

Minor keloids that failed to improve within 8–12 weeks with silicone gel or sheeting and intralesional steroids or major keloids that failed to improve with intralesional corticosteroids and 5-FU may be treated with laser therapy [117]. Thick keloids show minimal improvement with 585 nm PDL treatments and better clinical results are reported when PDL is used in combination with intralesional corticosteroids or 5-FU injections or when fractional CO_2_ laser is used [24]. PDL can be beneficial in attenuating pain and pruritis associated with keloids, but because PDL specifically targets scar erythema and not scar thickness, it is not recommended as a monotherapy for keloid management [127,128].

Contact and intralesional cryotherapy are other modalities with reported benefits for the treatment of keloids (Figure 10) [105,115,129]. Cryotherapy freezes the lesion using liquid nitrogen or argon gas and intralesional cryotherapy targets the keloid from the inside [129,130]. Contact cryotherapy is beneficial in combination with other modalities like intralesional corticosteroids, but patients may require up to 20 treatments [131]. Intralesional cryotherapy has been reported as more efficacious and requires fewer treatments compared to contact cryotherapy [129]. A comprehensive review of eight studies reported an average scar volume reduction of 51–63% and average scar recurrence rates of 0–24% in keloids treated with intralesional cryotherapy [130]. Furthermore, intralesional cryotherapy is a good alternative for pediatric patients who cannot undergo adjuvant radiotherapy following the surgical excision of keloids [130].

As with hypertrophic scars, when the above therapies have been utilized for at least 12 months for primary wound treatment and the keloid failed to resolve, surgical re-excision is recommended [4]. We recommend following the same algorithm regarding minimizing wound tension and utilizing silicone gel or sheets, scar massaging, and pressure therapy in the immediate post-operative period. Intralesional corticosteroids should be used intra-operatively and post-operatively as they demonstrate great effectiveness in pediatric patients [107].

## 6. Limitations and Future Directions

The present review is limited by the dearth of available articles describing scar management specifically for the pediatric population. While the review included these articles, the majority of the management modalities were extrapolated from adult cases. Additionally, many of the articles reviewed did not distinguish pediatric scar management across all pediatric ages, which may have an effect on healing. Another limitation of this study is that the included literature contains data on acne, post-operative, post-burn, and post-traumatic scars. Our proposed evidence-based algorithm is organized based on the appearance and pathological healing of the scar, not the underlying etiology of scar formation. As such, we cannot conclude which etiologies of scar formation may benefit most from our proposed algorithm. Future research should compare the cosmetic results of following our proposed scar management algorithm for acne, post-operative, post-burn, and post-traumatic scars in pediatric patients across various age groups.

## 7. Conclusions

Pediatric scars can have a huge social and psychological impact on patients and families. An algorithmic approach should be taken to minimize the chance of scar formation and effectively treat scars once they are formed (Figure 11). While evidence for pediatric-specific scar optimization approaches is limited, general principles can be applied across all age groups. Proper scar prevention should begin with careful incision planning parallel to tension lines and minimum tension wound closure. In the post-operative period, conservative therapies including silicone gel or sheets and massage therapy should be prioritized for 2–6 months. If the immature scar remains erythematous, PDL is recommended and if the immature scar remains thick, ablative CO_2_ laser is recommended. As the mature scar forms, it is important to identify pathological scar formation and begin the respective treatments immediately for atrophic scars, hyperpigmentation, hypertrophic scars, and keloids.

## Figures and Tables

**Figure 1 medicina-61-00553-f001:**
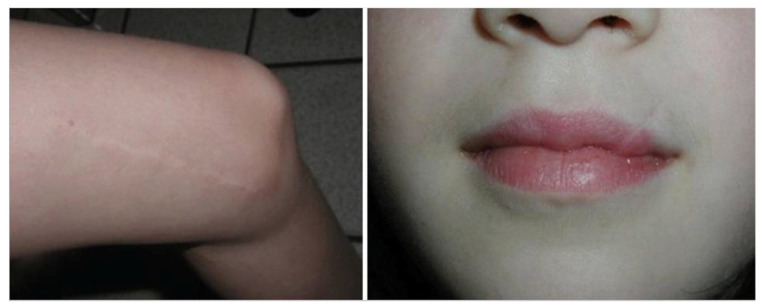
“Ideal” scars, healed as a thin pencil line with minimal color difference between scar and surrounding skin. Source: Textbook on Scar Management: State of the Art Management and Emerging Technologies, Chapter 46. Ref. [2].

**Figure 2 medicina-61-00553-f002:**
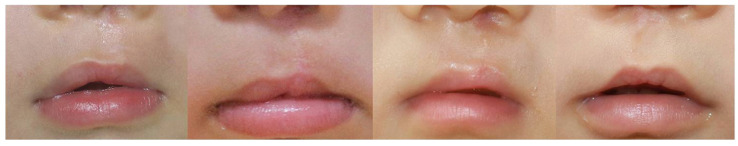
Babies from the study group (silicone gel group) showing similar quality scars to the control group (silicone sheet group). Source: Clinical Evaluation of Silicone Gel in the Treatment of Cleft Lip Scars. Ref. [15].

**Figure 3 medicina-61-00553-f003:**
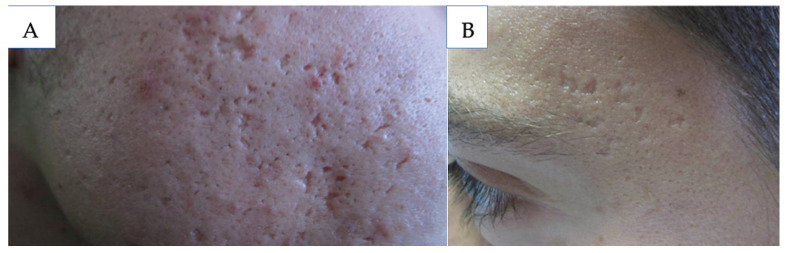
(**A**) Icepick scars; (**B**) boxcar scars. Source: Acne Scars: Pathogenesis, Classification and Treatment. Ref. [44].

**Figure 4 medicina-61-00553-f004:**
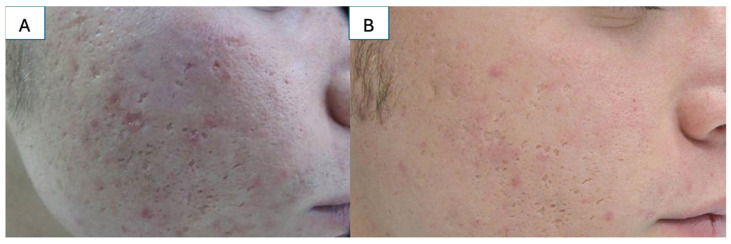
(**A**) TCA Cross: patient before the treatment; (**B**) TCA Cross: patient after the treatment. Source: Acne Scars: Pathogenesis, Classification and Treatment. Ref. [44].

**Figure 5 medicina-61-00553-f005:**
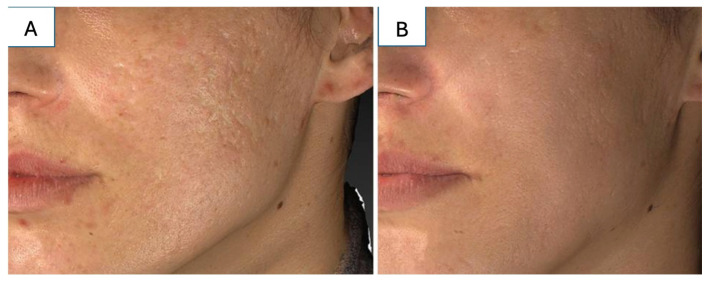
(**A**) Atrophic scarring on the face; (**B**) the same patient 6 months after fractional CO_2_ laser treatment. Source: Textbook on Scar Management: State of the Art Management and Emerging Technologies, Chapter 41. Ref. [57].

**Figure 6 medicina-61-00553-f006:**
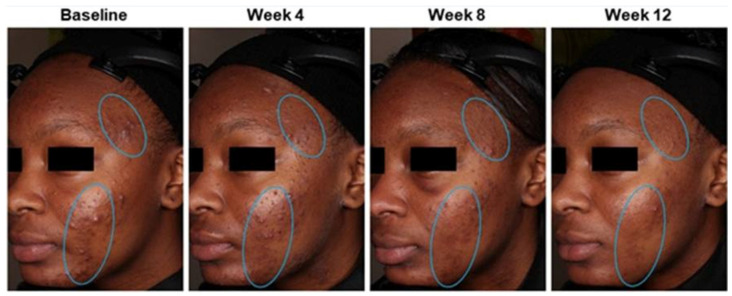
Acne hyperpigmentation (indicated by circled areas) improvements in a 15-year-old who was treated with tazarotene 0.045% lotion once daily for 12 weeks. Source: Effects of Topical Retinoids on Acne and Post-inflammatory Hyperpigmentation in Patients with Skin of Color: A Clinical Review and Implications for Practice. Ref. [82].

**Figure 7 medicina-61-00553-f007:**
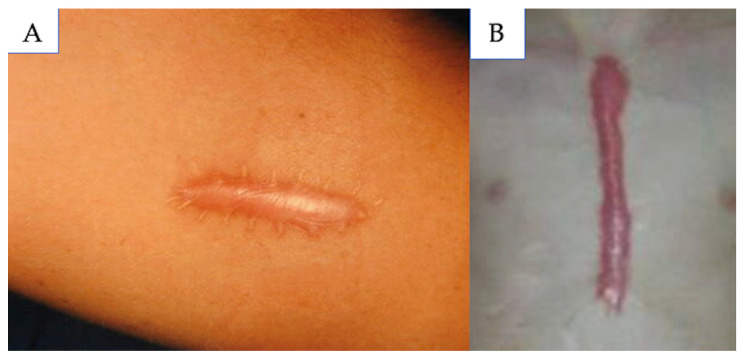
(**A**) Hypertrophic scar; (**B**) chest hypertrophic scar. Source: Textbook on Scar Management: State of the Art Management and Emerging Technologies, Chapters 9 and 46. Refs. [2,41].

**Figure 8 medicina-61-00553-f008:**
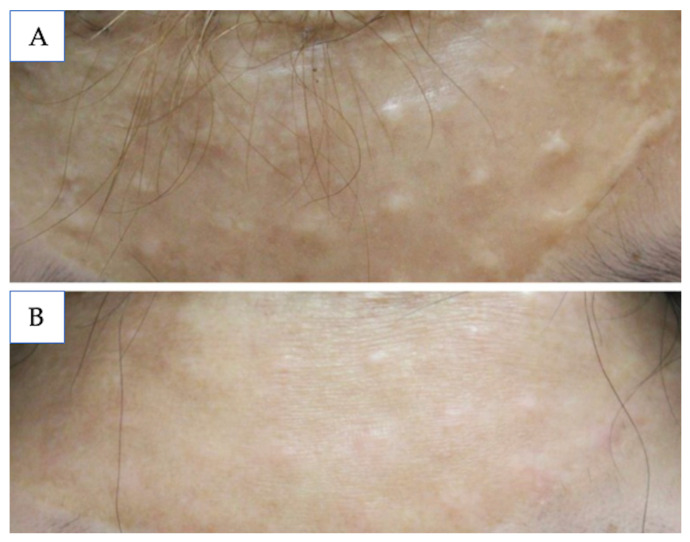
(**A**) A 16-year-old woman with hypertrophic scars after facial reconstruction surgery. (**B**) The patient underwent five sessions of fractional CO_2_ laser combined with the topical application of tramcinolone aetonide suspension of 10 mg/mL. Source: Textbook on Scar Management: State of the Art Management and Emerging Technologies, Chapter 50. Ref. [114].

**Figure 9 medicina-61-00553-f009:**
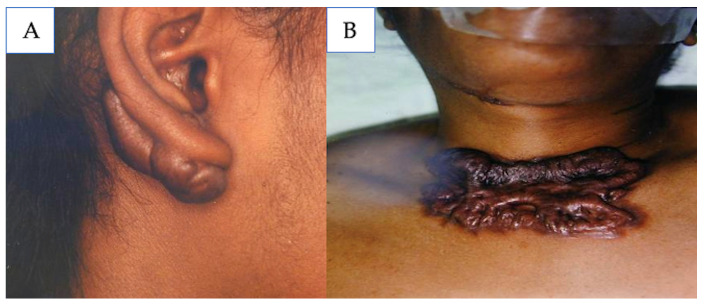
(**A**) Earlobe keloid; (**B**) chest keloid. Source: Textbook on Scar Management: State of the Art Management and Emerging Technologies, Chapter 9. Ref. [41].

**Figure 10 medicina-61-00553-f010:**
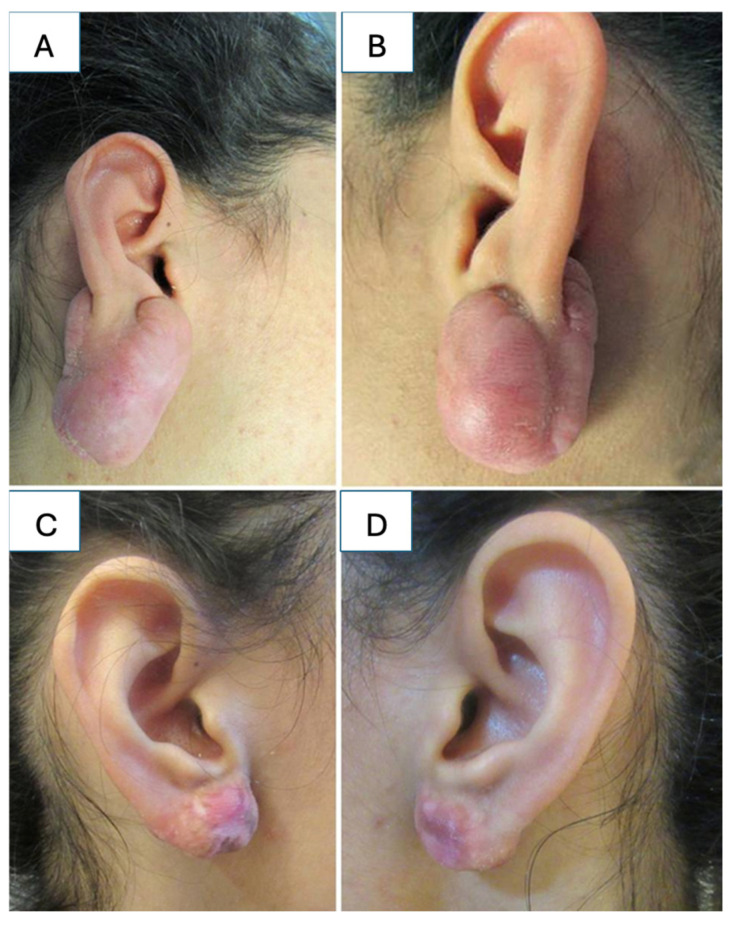
(**A**,**B**) Earlobe keloids; (**C**,**D**) post-operative view 6 years following a single cryosession of intralesional cryosurgery demonstrating complete involution of the scars with no distortion of the lobules and without hypopigmentation or recurrence. Source: Textbook on Scar Management: State of the Art Management and Emerging Technologies, Chapter 28. Ref. [129].

**Figure 11 medicina-61-00553-f011:**
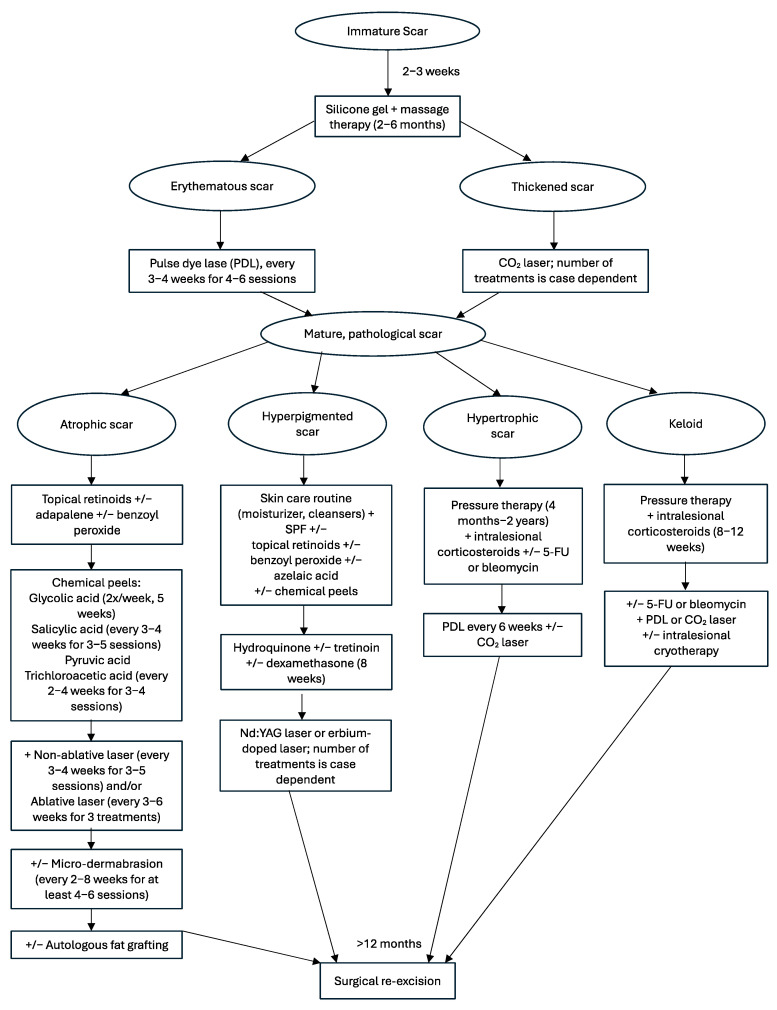
Pediatric scar management algorithm.

## Data Availability

Available upon request from the corresponding author.

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
