# Peer review of "Scar Management in Pediatric Patients"

_medicina, 2025, doi:10.3390/medicina61040553_

Round 1
Reviewer 1 Report
Comments and Suggestions for Authors
The authors present a literature review examining scar management among paediatric burn survivors. This is a timely and interesting work considering the impact of burns on children and the long-term nature of post-burn scars. Please see the comments below to help strengthen the manuscript:
- Several aspects of the study are well raised albeit it remains unclear what type of review this is to judge the overall conduct of the review. The authors move from the introduction to "managing the scars before it forms". How did the authors identify this? What were the search types? How were the data synthesised? Please revise this part to strengthen the overall content.
2. The limitations are not highlighted as well. Please consider adding this
Reviewer 2 Report
Comments and Suggestions for Authors
The article provides a good summary of the treatment algorithm for childhood scar tissue. Figure 1 excellently summarizes the treatment options for different types and stages of scars. The structure of the paper is good. The quality of the paper would be significantly improved with an illustrated presentation, and it would be important to discuss the results achieved with various treatment methods before and after the treatments. I support the publication of the article, but with the inclusion of an illustrated presentation.
Reviewer 3 Report
Comments and Suggestions for Authors
The article is well-documented and carefully written. However, there are some important issues to be addressed.
The introduction should state more carefully that the review analyses post-acne scars and post-operative scars, with less emphasis on post-traumatic and post-burn scars. This could be stated as a limitation or further development of the study.
The material and method section, although mentioned in the abstract, is not developed within the main manuscript, and it does not mention the criteria and the time span of the review. Correlating this with the references, they should be updated or refined, some of them are more the 20-25 years old.
In the atrophic and hyperpigmented sections, there are no statements with regard to whether those treatment options are approved or safe in children, as it is mentioned in the following sections. Since it is a review dedicated to paediatric patients, it should clarify very thoroughly which options are safe and approved for children, and which are only off-label uses or based on adult reports.
There are also some punctual rephrasing revisions that are needed
„Mature scars are typically greater than one year old” – older than
„Early primary wound closure is critical to induce wound epithelization as soon as possible because wound epithelialization delayed beyond 10-14 days increases the risk for the formation of hypertrophic scars.” – repetition and misspelling of the bolded words
„Management of keloids is similar to the management of hypertrophic scars, but should be more aggressive as keloids are less likely to regress and can continue to grow for a longer than hypertrophic scars” incomplete sentence
Finally, the conclusions should ne adjusted in accordance with the changes made to the main manuscript.
Round 2
Reviewer 1 Report
Comments and Suggestions for Authors
Thanks to the authors for addressing the comments raised. It will be helpful if the authors indicate that the study is a narrative review just to keep authors on track.
Reviewer 3 Report
Comments and Suggestions for Authors
The authors have revised the manuscript and made most of the suggested changes. The paper has now an improved structure.
However, there is an important point that has not yet been addressed:
In my previous review I suggested to the authors to mention in the method section (newly introduced) the time span of the review. I still did not find this information in the manuscript.
Since the aim of the review paper is to propose an algorithm for scar management in pediatric patient, this algorithm should be based on the most recent research and data. The presence of 22 references that are more than 20 years old (time range: 1987-2005), some of them being clustered, implies a risk of providing to the readers information that might be outdated. I am suggesting again an update of references, and elimination of the old ones (3, 15, 20, 23, 31-32, 50-51, 53-54, 87-90, 95, 112-113, 121, 125-126, 128, 138-139). The topics related with these references are iontophoresis, PIH at the beginning, hypertrophic scars, use of 5-FU and PDL.
